# Isolated Proximal Median Neuropathy after Aortic Dissection Repair: Case Report

**DOI:** 10.3390/medicina58050622

**Published:** 2022-04-29

**Authors:** Yen-Yu Chen, Chao-Chun Huang, Jia-Chi Wang, Po-Cheng Hsu

**Affiliations:** 1Department of Physical Medicine and Rehabilitation, Taipei Veterans General Hospital, Taipei 112, Taiwan; d29232003@gmail.com (Y.-Y.C.); jcwang0726@gmail.com (J.-C.W.); 2Department of Physical Medicine and Rehabilitation, Far Eastern Memorial Hospital, New Taipei City 220, Taiwan; baberuth19601036@gmail.com; 3School of Medicine, National Yang Ming Chiao Tung University, Taipei 112, Taiwan

**Keywords:** proximal median neuropathy, ultrasonography, electrodiagnosis, case report

## Abstract

Surgery-related isolated proximal median neuropathy is a rare complication. Brachial plexus injury is a possible complication after major cardiac surgery; however, isolated mononeuropathy is less frequently documented. We present an unusual case of isolated proximal median neuropathy after aortic dissection repair surgery in a 39-year-old man. Electrodiagnostic study and ultrasound examinations helped in localizing the lesion to the axillary region. Serial follow-ups showed improvement in electrodiagnostic parameters, which were compatible with clinical symptoms. Partial recovery was achieved at the seventh month follow-up. This case report aimed to increase awareness of nerve stretching during open heart surgery and demonstrate the diagnosis and clinical follow-up by concomitant use of electrodiagnostic and nerve ultrasound studies.

## 1. Introduction

Median mononeuropathy is not a rare condition; however, most lesions develop in the carpal tunnel. In a s study reviewing 4838 electrophysiological studies during a 5-year duration, approximately 0.2% of patients having proximal median neuropathy while 23% having median nerve neuropathy at carpal tunnel. [1]. The etiologies of proximal median neuropathy include trauma, external compression, venipuncture, tumors, and hematoma [2].

The median nerve, consisting of the C5 to T1 roots, is derived from the medial and lateral cords of the brachial plexus. It runs in the medial aspect of arm and passes between the two heads of the pronator teres, and the main trunk of the median nerve further enters the carpal tunnel while branching to the anterior interosseous nerve (AIN). The median nerve innervates most flexor muscles of the forearm and provides sensory innervation to the volar side of the first three and a half fingers [3]. The symptoms of proximal median neuropathy depend on the site of the lesion. A lesion above the elbow affects all flexor and pronator muscles innervated by the median nerve, which presents as an inability to flex the first three digits and perform forearm pronation. Moreover, numbness of the thenar volar site indicates that the lesion is above the carpal tunnel because of palmar cutaneous nerve branching before entrance into the tunnel [2].

Upper limb neuropathy is a common complication of thoracotomy. The prevalence of brachial plexus injury (BPI) after open chest surgery is between 2% and 38% [4,5]. The main open heart surgical procedure is coronary artery bypass grafting following valve replacement surgery [6]. However, data on the prevalence of BPI after aortic dissection repair surgery and isolated proximal median neuropathy are lacking. We present a rare case of isolated proximal median neuropathy after aortic dissection repair.

## 2. Case Presentation

A 39-year-old man without previous metabolic diseases or genetic disorders associated with predisposition to neuropathies presented with left forearm and hand weakness, pain, and numbness for a month. All the symptoms appeared after open chest aortic dissection repair surgery. The surgery took 8 h, and the patient was placed in the supine position with the left arm extended. The repair was performed smoothly; however, immediately after surgery, the patient had throbbing pain and numbness over the forearm and fingers, especially in the thenar region. He also found himself unable to pronate the forearm or flex the first three digits of the left hand. The symptoms were not relieved by oral analgesic administration; thus, he visited our outpatient department.

Physical examination revealed left wrist flexor weakness, and the muscle power was graded based on the Medical Research Council (MRC) scale as 5 in right wrist flexion but 0 in left wrist flexion. The tendon reflexes over the left biceps brachii, triceps, and brachioradialis were grade 2+ based on the National Institute of Neurological Disorders and Stroke muscle stretch reflex scale. Moreover, the patient was unable to make the “OK sign,” indicating the inability to move the distal interphalangeal joints with a compensatory hyperextension. Sensory examination showed more prominent hypoesthesia in the palmar aspect of the first three fingers of the left side compared to that of the fingers of the right side. A clinical impression of proximal median neuropathy was made.

Electrodiagnostic studies were performed at the first, fourth, and seventh months after injury. At the first month after injury, the nerve conduction study showed markedly decreasing amplitude and conduction velocity over the left median compound motor action potential, absence of F response while recording at the abductor pollicis brevis (APB), and absence of left median sensory nerve action potential. Right median nerve and bilateral ulnar and radial nerve conduction studies yielded results within normal limits. Needle electromyography (EMG) showed active denervations over the left flexor carpi radialis (FCR) and APB with decreased recruitment (Table 1). These results implied that the site of proximal median nerve injury was located before branching to FCR.

Ultrasound examination was performed simultaneously with electrodiagnostic examination to localize the lesion. A long segment of hypoechoic swelling of the left median nerve was noted from the axillary level until branching to AIN, by side-to-side comparison (Figure 1A,B). Ultrasound-guided hydrodissection of the left median nerve with corticosteroid administration over whole arm from axillary level to elbow (Figure 1C) was performed at the first month after injury after electrodiagnostic examination. The injection was performed twice with a two-week interval. A serial rehabilitation program, two times per week, including neuromuscular electrical stimulation and left forearm flexor strengthening exercises, was established.

At the fourth and seventh months after injury, electrodiagnostic follow-up was arranged, which showed a gradual decrease in spontaneous activity and development of polyphasic waves over the left APB and FCR (Table 1). At the seventh month after injury, the patient reported sporadic numbness, and the muscle strength of left wrist flexors improved to MRC 3. After a year of rehabilitation, the muscle strength of left wrist flexors recovered to the premorbid condition, but with remaining neuropathic pain.

## 3. Discussion

Iatrogenic peripheral neuropathies are usually caused by transection, stretching, compression, radiation, and puncture by injection [7]. Besides the carpal tunnel, the ligament of Struthers, bicipital bursa, and Gantzer’s muscle are all possible forearm areas in which the median nerve may be compressed. The most common median neuropathies other than carpal tunnel syndrome are AIN and pronator syndrome [8]. However, median neuropathy at the arm is far less frequently recorded. Proximal median neuropathy after aortic dissection repair is a rare complication. To the best of our knowledge, there has been no previous report of this condition in the literature.

Considering the patient’s history, the symptoms developed after aortic repair surgery. Thus, the interventions during surgery, such as sternotomy, compression, and stretching in a specific position, may be possible etiologies contributing to neuropathy. According to previous studies, the brachial plexus, particularly the lower trunk, is exposed to forcible stretching during sternotomy [4]. Other possible related mechanisms for neuropathy include arm placement during surgery, concomitant first rib fracture, compression and traction of the brachial plexus, cannulation of the internal jugular vein, and increased bypass duration [6].

However, physical examination and electrophysiological testing in the present case showed isolated median neuropathy before branching to the FCR. Traction during sternotomy or internal jugular vein cannulation affecting the brachial plexus at the trunk level is less likely, and no first rib fracture was detected. Positioning during surgery is the most possible mechanism. The standard position during aortic dissection repair is shoulder depression and external rotation with elbow slight extension, unlike the common position that causes brachial plexus stretching injury while the patients’ arms are placed above the head with exaggerated abduction. Although the median nerve is stretched mostly when the shoulder is in 90° abduction with the elbow and wrist extended by the upper limb nerve tension test [9], the standard aortic dissection repair position may partially elevate the median nerve tension in some cases especially in prolonged surgery. In the previous study, with the elbow in full extension and wrist in neutral position, the shoulder position influenced the tension of the proximal median nerve more than that of the distal part [9], which caused an unusual proximal median neuropathy at the arm level.

Electrophysiological testing and ultrasonography are helpful tools for localizing the lesion in neuropathy. Our case showed decreasing amplitude and conduction velocity when stimulating the median nerve at the elbow and wrist with normal nerve conduction studies over the radial and ulnar nerves, which implies that the influence was on the median nerve rather than on the brachial plexus. The needle EMG suggested that the lesion was an isolated median neuropathy before branching to the FCR and well demonstrated the time course of neuropathy from the acute phase to the reinnervation phase. Nerve ultrasonography further confirmed the lesion at the axilla and showed the classical pattern of neuropathy, which is a hypoechoic and enlarged nerve at the axilla compared with that at the other side. Though MRI is widely used in evaluating brachial plexus injury, ultrasonography has advantages in focal resolution of individual peripheral nerve, accessibility, and without contraindication [10,11]. Meanwhile, ultrasonography can simultaneously being a guided tool for injection intervention.

By ultrasonography and serial EMG studies, we observed that the patient showed axonotmesis according to the Seddon and Sunderland classification initially [12], as per which the prognosis was relatively poor whether surgery was performed or not in a previous case series [1]. After discussion with the patient, conservative treatment with rehabilitation and ultrasound-guided hydrodissection of the injured site were performed. The patient reported improvement to satisfactory strength and sensory conditions, which was further ascertained by EMG at the seventh month follow-up.

## 4. Conclusions

Our case demonstrates the possibility of isolated proximal median neuropathy after aortic dissection repair. During open heart surgery, care should be taken not to tighten the shoulder, elbow, and wrist, since it may lead to nerve stretching. Once numbness and weakness develop, concomitant electrodiagnostic studies and nerve ultrasonography are useful for locating the lesion and determining the prognosis, which can indicate whether a conservative or invasive treatment should be applied.

## Figures and Tables

**Figure 1 medicina-58-00622-f001:**
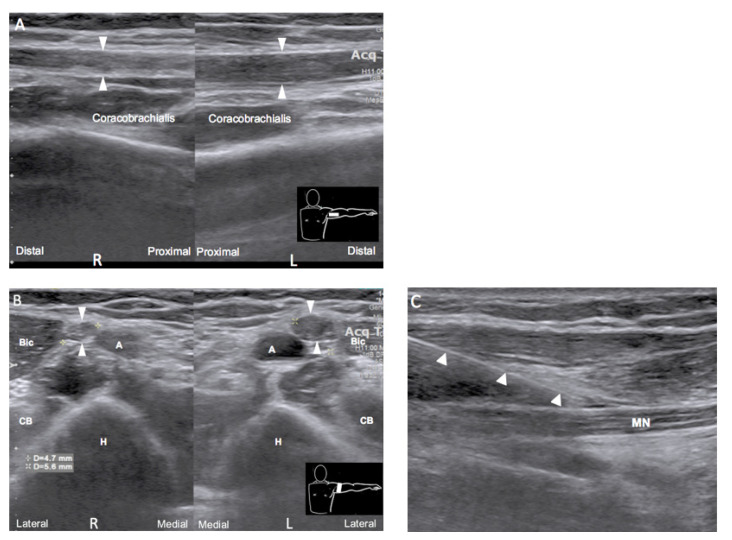
Comparison of the bilateral median nerve at the axillary level in longitudinal view (**A**) and transverse view (**B**) showing hypoechoic and enlarged caliber of the left median nerve. Arrowhead: median nerve. A, axillary artery; H, humerus; Bic, biceps muscle; CB, coracobrachialis muscle. (**C**) Hydrodissection of the left median nerve at the mid-arm level. Arrowhead: needle; MN: median nerve.

**Table 1 medicina-58-00622-t001:** Serial electrodiagnostic studies post-injury.

	1st Month			4th Month			7th Month		
	Amp	Lat	V	Amp	Lat	V	Amp	Lat	V
Motor Proximal/distal									
L Median	0.1/0.4	4.4	38.1	0.1/0.1	4.12	51.9	0.2/0.2	5	34
L Ulnar	6.9/7.2	2.7	52.8	7.9/5.7	2.42	65.7	5.6/5.9	2.7	53
Sensory									
L Median	-	-	-				4.1	6	17
L Ulnar	21.7	2.3	60.9				24.8	2.5	42
	ASA	MUAP	RP	ASA	MUAP	RP	ASA	MUAP	RP
L APB	++	Few	Discrete	+	Poly	Moderate reduced	-	Poly	Mild reduced
L FCR	++	Few	Discrete	+	Poly	Moderate reduced	-	Poly	Mild reduced

L, left; Amp, amplitude; Lat, latency; V, velocity; ASA, abnormal spontaneous activity; APB, abductor pollicis brevis; FCR, flexor carpi radialis; MUAP, motor unit action potential; RP, recruitment patterns; Poly, polymorphic motor unit action potential.

## Data Availability

The data presented in this study are available on request from the corresponding author. The data are not publicly available owing to privacy.

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
