# Peer review of "Isolated Proximal Median Neuropathy after Aortic Dissection Repair: Case Report"

_medicina, 2022, doi:10.3390/medicina58050622_

Round 1

Reviewer 1 Report

  1. Was there any further follow-up of the patient? Why was follow-up discontinued at 7 months post-op in the view of incomplete nerve function recovery and the fact that nerve regeneration is known to continue for up to 2 years after an injury?

  2. Introduction: The context in which the data from the article by Gross and Jones is cited is unclear. Please, present some context of the patient group studied by these authors.

  3. Would you recommend the use of MRI in such nerve injuries? Please, include your answer in the Discussion.

  4. When did the treatment process take place (months/year)?

  5. How long after the injury was hydrodissection performed?

  6. At what length of the nerve was the hydrodissection conducted: was it from the axilla to the mid-arm level or locally at the mid-arm level?

  7. How long did the rehabilitation after hydrodissection last?

  8. Case Presentation, line 3 from the top: Did the nerve injury symptoms exacerbate after open chest surgery (which means they were present before the surgery but got worse) or did they appear for the first time after surgery?

  9. Case Presentation, line 7 from the top: Did the authors mean fingers instead of finger?

  10. What do you mean by take-off of FCR? Please, explain or rephrase.

  11. Case Presentation: The word „injury” is missing in the following sentence „ These results implied that the proximal median nerve injury was above the take-off of FCR”.

Reviewer 2 Report

as this is rare complication but only one case, this can be accepted as letter to editor to just record this complication in literature.

it can not be accepted as case report but only as letter 
